# Frictional Heating during Braking of the C/C Composite Disc

**DOI:** 10.3390/ma13122691

**Published:** 2020-06-12

**Authors:** Aleksander Yevtushenko, Michal Kuciej, Katarzyna Topczewska

**Affiliations:** Faculty of Mechanical Engineering, Bialystok University of Technology (BUT), 45C Wiejska Street, 15-351 Bialystok, Poland; a.yevtushenko@pb.edu.pl (A.Y.); k.topczewska@pb.edu.pl (K.T.)

**Keywords:** braking, frictional heating, temperature, C/C composite, multi-disc brake

## Abstract

An analytical model to determine temperature in a single brake disc of multi-disc system is proposed. The model considers the convective cooling on the lateral surfaces of the disc and structure of composite friction material. Calculations were carried out for a disc made of carbon friction composites material Termar-ADF. The influence of heat transfer with environment, length of bundles with fibers, and concentration of fibers in composite on the temperature of the disc was investigated during single braking with constant deceleration.

## 1. Introduction

At the present time, high-speed and heavy-duty vehicles are extensively developed. These include high-speed rail and road transport, machines for military purposes, and, in particular, aircraft. For the successful operation of these categories of vehicles, appropriate braking devices are required that ensure their sufficient maneuverability. To design brakes, it is necessary to fulfill a complex of requirements (ensuring high braking torque and wear resistance, absence of overheating, stability of characteristics, etc.), which is associated with the choice of friction materials, design of the brake unit, bench, and field tests [1].

The braking system of modern passenger as well as military aircraft consists of package of fixed and rotating friction discs located inside the wheel. After switching on the brake, the special pistons compress package, the front surfaces of discs are in contact, and the process of friction braking begins. While the kinetic energy of the aircraft is absorbed, the friction between fixed and rotating discs causes that they are heated up even to 1500 °C, and temperatures of the friction surfaces reach up to 3000 °C. Therefore, the disc’s materials must be resistant enough to thermal shock [2].

It is known that the friction polymer materials start to degrade at the temperature of 230 °C, and the degree of degradation increases with temperature within the range 269–400 °C [3,4]. The degradation of the polymer materials causes brake fade phenomena, where the coefficient of friction is reduced with increasing temperature [5]. The high temperature decreases the yield strength and leads to changes in the wear mechanism and the real contact configuration [6]. These phenomena could increase the wear rate of the brake friction material [7]. Therefore, to improve the working life of aircraft brakes, there is an increasing use of carbon friction composites materials (CFCM) that differ in the type of reinforcement, the method of forming the friction layer, the variations of technological heat treatment, etc. For such a material, a certain degree of wear intensity is typical, but, in general, CFCM has significant advantages over friction metal ceramics and plastics. CFCM materials are characterized by less tendency to increase wear with rise in temperature, as well as smaller sensitivity to changes in the specific pressure and sliding speed [8,9,10].

Carbon-carbon (C/C) composite materials are used at heavy working conditions, under which the temperature of the surface of friction can exceed 1500 °C and the bulk temperature 800 °C. CFCM such as Carbenix-4000 (Honeywell International Inc., Charlotte, NY, USA), Sepcarb (Safran Landing Systems, Vélizy, France), and Termar (PJSC Aviation Corporation "Rubin”, Russia) have a low density and thermal linear expansion coefficient, high thermal conductivity, energy-absorption capability, stable coefficient of friction, and retain mechanical properties up to 2400–2800 °C [11,12,13,14]. Carbon composite brake discs are lighter, economical, and have excellent high energy friction characteristics. Two basic advantages of C/C materials at high temperatures—the heat capacity and strength (higher by 2.5 and 2 times, respectively, than steel)—result in a 40% weight savings [2]. These remain unaffected by thermal shocks and mechanical fatigue [15]. Due to such a specific combination of the properties, C/C composites are more widely used for producing brakes subjected to heavy loads and extreme application conditions, and can be considered as the basic construction material for high-temperature applications, e.g., in aerospace and aircraft industry brake systems. Currently, roughly 60% of produced C/C composites accounts for aircraft braking discs [16,17]. Carbon composites are also widely used for the production of automobile brakes and could potentially be used in brakes for high-speed trains [18,19].

Depending on the purpose, various modifications of CFCM of the Termar type are used, which have different thermal, mechanical, and friction-wear properties. The discrete-reinforced transversely isotropic friction C/C Termar-ADF is a material with the formation of the original press package by the random arrangement of long fibers with use of special installation. Termar-ADF is fabricated using a pitch matrix reinforced by high modulus fibers with the average length being 20 mm. Fibrous materials are used for two cycles of impregnation with thermoplastic pitch using liquid-phase pyrolysis at 600 °C with pressure up to 100 MPa. Each cycle of impregnation is followed by carbonization at 1200 °C. The final heat treatment is performed at 2000 °C for 1 h [20].

The calculation of the temperature regime and wear characteristics of the brake is the most important step in the design of the brake wheel as a whole. One of the most loaded elements of the brake, receiving greater thermal and mechanical loads, are the discs. Numerous natural tests and studies on the brakes have shown that the main factor determining the wear resistance of the C/C material of the discs, and hence the life of the product as a whole, is the temperature. Thus, the determination of the temperature field of the disc is necessary for predicting the durability of the brake. The result of test for heat pulsing friction of brand Termar-ADF carbon frictional composite material on the multipurpose friction machine IM-58-T2 is presented in [21]. The effect of the reinforcement type and the heat treatment conditions of the CFCM on their friction coefficient, wear rate, and the stability indicator of the friction coefficient is studied in [22].

The complex methodology for the design and manufacture of brakes for air wheels is presented in [23,24]. It includes the theoretical and experimental technique of load and thermal calculation of friction couples and modeling of their work under the conditions of varying external operational factors. The use of that methodology allows optimizing the design of the brake at the given dimensions according to parameters such as the braking torque and its stability, the duration and way of braking, the brake resource according to the wear factor, and volumetric and surface temperature. One of the most important components of the thermal calculation of the brake is the solution of the thermal problem of friction—the boundary-value problem of heat conductivity for a friction element heated by a frictional heat flow on the working surface and cooled on its free surfaces. Reviews of analytical and numerical solutions of this problem are presented in [25,26]. It is noted that most authors have considered the case of a homogeneous disc material with adiabatic free surfaces. An attempt to take into account the convective cooling of the lateral surfaces of a homogeneous disc during braking with constant deceleration is undertaken in [27]. Unfortunately, it presents formulas for determining the temperature only on the friction surface of the disc. The generalization of such solution in the case of a disc made of composite C/C material is executed in [28]. The solution is obtained in the form of the Duhamel integral, which is useful in practical applications.

For a multi-disc brake pair made of composite C/C materials, a numerical finite element method (FEM) solution of thermal problem of friction is obtained in [29], and an analytical solution is presented in [30]. That solutions are obtained considering effective properties of disc composite materials. This article provides analytical solution to the thermal problem of friction, considering the structure of the composite material. In detail, the main aim of the article is to obtain an exact solution to the thermal problem of friction, which simulates the process of frictional heating of an individual disc of a multi-disc brake during single braking. It is assumed that the disc is made of composite C/C material containing carbon fibers, collected in bundles, which are located in planes parallel to the surface of friction. At the first stage, an exact solution of the corresponding thermal problem of friction for an orthotropic disc is obtained, considering the effective thermal conductivity coefficients of the material in the axial and radial directions, as well as convective cooling of the lateral surfaces. In the second stage, the procedure for determining the above-mentioned effective properties of the composite at three levels (micro, meso, and macro) is proposed. Based on the obtained solution, the distribution of a transient temperature field in a disc made of Termar-ADF is investigated.

## 2. Statement to the Problem

Multi-disc aircraft braking systems are subjected to heavy mechanical and thermal loads during braking. Typically, such systems are a package of identical discs. A scheme of such multi-disc system and the calculation diagram of the considered problem are presented in Figure 1. A microscopic view of the real structure of such C/C composite can be found in [15,20] and sample photo of single brake disc made of Termar-ADF in [31].

It is assumed that all friction discs have the same internal radius $r_{1}$, external radius $r_{2}$, thickness 2d, and are made of orthotropic material with different thermal conductivities $K_{r}$ and $K_{z}$ in radial r and axial z directions, respectively. Each of the two outer fixed discs has one friction surface, while the inner rotating and fixed discs have two friction surfaces, and thus they heat up more. Since all the discs are made of the same material, the temperature fields in the internal discs are approximately the same, and their distribution over the thickness of the disc is symmetrical relative to its middle surface. Therefore, the corresponding thermal problem of friction can be formulated not for a whole disc with a thickness of 2d, but only for its half with a thickness of d, assuming that the friction surface z=0 is heated by frictional heat flow with the intensity q and the plane of symmetry z=d is adiabatic. Since the thermal conductivity of CFCM in the circumferential direction parallel to the working surfaces of the disc is much higher than in the axial direction z of action of the heat flow, the distribution of the temperature field on each surface 0<z<d is uniform. Lateral surfaces of the discs $r=r_{1}$ and $r=r_{2}$
0≤z≤2d are cooled convectively with constant coefficient of heat transfer h. The mutual overlap of the contacting elements is full, so the nominal contact area $A_{a}=\pi (r_{2}^{2}-r_{1}^{2})$ remains unchanged during braking.

Taking the above-mentioned assumptions into account, distribution of the temperature T across the disc thickness 0≤z≤d at a fixed moment of time $0\leq t\leq t_{s}$, we may find from solution to the following boundary-value problem of heat conduction [28]:(1)$$\frac{\partial^{2}T(z,t)}{\partial z^{2}}-\frac{2h^{*}}{K_{z}l}[T(z,t)-T_{a}]=\frac{1}{k}\frac{\partial T(z,t)}{\partial t},\;0<z<d,\;0<t\leq t_{s}$$
(2)$${K_{z}\frac{\partial T(z,t)}{\partial z}|}_{z=0}=-q(t),\;0<t\leq t_{s}$$
(3)$${K_{z}\frac{\partial T(z,t)}{\partial z}|}_{z=d}=0,\;0<t\leq t_{s}$$
(4)$$T(z,0)=T_{a},\;0\leq z\leq d$$
where
(5)$$q(t)=q_{0}q^{*}(t),\;0\leq t\leq t_{s}$$
(6)$$q_{0}=fp_{0}V_{0},\;V_{0}=\omega_{0}r_{m},\;r_{m}=0.5(r_{1}+r_{2}),$$
(7)$$q^{*}(t)=1-\frac{t}{t_{s}},\;t_{s}=\frac{W_{0}}{q_{0}A_{a}}$$
(8)$$h^{*}={\left(\frac{1}{h}+\frac{r_{2}-r_{1}}{2K_{r}}\right)}^{-1}$$
(9)$$k=\frac{K_{z}}{\rho c}$$

$T_{a}$ is the initial temperature of the disc; t is the time; $t_{s}$ is the time of braking; f is the coefficient of friction; ρ is the density; c is the specific heat of material; $p_{0}$, $\omega_{0}$, and $W_{0}$ are respectively initial values of contact pressure, angular velocity, and kinetic energy.

Introducing the dimensionless variables and parameters:(10)$$\zeta =\frac{z}{d},\;\tau =\frac{kt}{d^{2}},\;\tau_{s}=\frac{kt_{s}}{d^{2}},\;Bi=\frac{2h^{*}d^{2}}{K_{z}(r_{2}-r_{1})},\;T_{0}=\frac{q_{0}d}{K_{z}},\;T_{}^{*}=\frac{T-T_{a}}{T_{0}}$$

We write the problem in Equations (1)–(4) as:(11)$$\frac{\partial^{2}T^{*}(\zeta ,\tau )}{\partial \zeta^{2}}-BiT^{*}(\zeta ,\tau )=\frac{\partial T^{*}(\zeta ,\tau )}{\partial \tau },\;0<\zeta <1,\;0<\tau \leq \tau_{s}$$
(12)$${\frac{\partial T^{*}(\zeta ,\tau )}{\partial \zeta }|}_{\zeta =0}=-q^{*}(\tau ),\;0<\tau \leq \tau_{s}$$
(13)$${\frac{\partial T^{*}(\zeta ,\tau )}{\partial \zeta }|}_{\zeta =1}=0,\;0<\tau \leq \tau_{s}$$
(14)$$T^{*}(\zeta ,0)=0,\;0\leq \zeta \leq 1$$
where
(15)$$q^{*}(\tau )=1-\frac{\tau }{\tau_{s}},\;0\leq \tau \leq \tau_{s}$$

It should be noted that the adoption of the linear model in Equations (1)–(9) to determine the temperature field in the brake disc, imposes some restrictions on the possibility of its use in medium and heavy modes of operation of the brake, when the surface temperature reaches values above 450 and 750 °C, respectively [3]. It is known that at such temperatures the coefficient of friction and the thermo-physical properties of the material may change the initial values. Application of the analytical models in such cases may involve the use of the friction coefficient averaged during braking; this approach has been used in the following numerical analysis. Thermal sensitivity of a material is usually taken into account by converting in its solution the value of the coefficient of thermal conductivity at room temperature to its value at the average volume temperature of the disc during braking. Of course, this approach is approximate, but gives no less sufficiently good compliance of the maximum temperature values with relevant experimental data [21]. To consider the thermal sensitivity of the above-mentioned characteristics, at each time step during braking, requires the development of appropriate nonlinear models, and the usage of numerical methods to solve them [29].

## 3. Solution to the Problem

By replacing [32]
(16)$$T^{*}(\zeta ,\tau )=\Theta (\zeta ,\tau )e^{-Bi\tau }$$
the boundary-value problem in Equations (11)–(15) is transformed into the form:(17)$$\frac{\partial^{2}\Theta (\zeta ,\tau )}{\partial \zeta^{2}}=\frac{\partial \Theta (\zeta ,\tau )}{\partial \tau },\;0<\zeta <1,\;0<\tau \leq \tau_{s}$$
(18)$${\frac{\partial \Theta (\zeta ,\tau )}{\partial \zeta }|}_{\zeta =0}=-q^{*}(\tau )e^{Bi\tau },\;0<\tau \leq \tau_{s}$$
(19)$${\frac{\partial \Theta (\zeta ,\tau )}{\partial \zeta }|}_{\zeta =1}=0,\;0<\tau \leq \tau_{s}$$
(20)Θ(ζ,0)=0, 0≤ζ≤1
where Θ(ζ,τ) is now the unknown function. Using the Duhamel’s theorem [33], the solution of the problem in Equations (17)–(20) can be expressed as:(21)$$\Theta (\zeta ,\tau )=-\int_{0}^{\tau }q^{*}(s)e^{Bi\;s}\frac{\partial }{\partial \tau }\Theta_{0}(\zeta ,\tau -s)ds,\;0\leq \zeta \leq 1,\;0\leq \tau \leq \tau_{s}$$
where function
(22)$$\Theta_{0}(\zeta ,\tau )=\zeta -\frac{1}{2}\zeta^{2}-\tau -\frac{1}{3}+2\sum_{n=1}^{\infty }\frac{e^{-\lambda_{n}^{2}\tau }}{\lambda_{n}^{2}}\cos(\lambda_{n}\zeta ),\;0\leq \zeta \leq 1,\;0\leq \tau \leq \tau_{s}$$
(23)$$\lambda_{n}=\pi \;n,\;n=1,2,\ldots \;.$$
is the solution to the problem in Equations (17)–(20) at $q^{*}(\tau )=1$ [34]. After integration on the right side of Equation (21) with functions $q^{*}(\tau )$ (Equation (15)) and $\Theta_{0}(\zeta ,\tau )$ (Equations (22) and (23)), we obtain:(24)$$\Theta (\zeta ,\tau )=J_{0}(\tau )+2\sum_{n=1}^{\infty }e^{-\lambda_{n}{}^{2}\tau }J_{n}(\tau )\cos(\lambda_{n}\zeta ),\;0\leq \zeta \leq 1,\;0\leq \tau \leq \tau_{s}$$
where
(25)$$J_{0}(\tau )=-\frac{1}{Bi}\left(1+\frac{1}{\tau_{s}Bi}\right)+\left(1-\frac{\tau }{\tau_{s}}+\frac{1}{\tau_{s}Bi}\right)\frac{e^{Bi\;\tau }}{Bi}$$
(26)$$J_{n}(\tau )=-\frac{1}{Bi_{n}}\left(1+\frac{1}{\tau_{s}Bi_{n}}\right)+\left(1-\frac{\tau }{\tau_{s}}+\frac{1}{\tau_{s}Bi_{n}}\right)\frac{e^{Bi_{n}\tau }}{Bi_{n}}$$
(27)$$Bi_{n}=Bi+\lambda_{n}^{2},\;n=1,2,\ldots .$$

Taking into account the sums of the series [35]:(28)$$\sum_{n=1}^{\infty }\frac{\cos(\lambda_{n}\zeta )}{Bi_{n}}=\frac{1}{2\sqrt{Bi}}\cos\mathrm{ech}(\sqrt{Bi})\mathrm{ch}[(1-\zeta )\sqrt{Bi}]-\frac{1}{2Bi}$$
(29)$$\begin{array}{ll} \sum_{n=1}^{\infty }\frac{\cos(\lambda_{n}\zeta )}{Bi_{n}^{2}} & =\frac{1}{4Bi}{\cos\mathrm{ech}}^{2}(\sqrt{Bi})\mathrm{ch}(\zeta \sqrt{Bi})+\\ & +\frac{1}{4Bi\sqrt{Bi}}\cos\mathrm{ech}(\sqrt{Bi})\{\mathrm{ch}[(1-\zeta )\sqrt{Bi}]+\zeta \sqrt{Bi}\;\mathrm{sh}[(1-\zeta )\sqrt{Bi}]\}-\frac{1}{2Bi^{2}} \end{array}$$

From Equations (16) and (24)–(27), we find the dimensionless temperature of the disc in the form:(30)$$\begin{array}{cc} T^{*}(\zeta ,\tau )= & \frac{\cos\mathrm{ech}(\sqrt{Bi})}{\sqrt{Bi}}\left\{\left(1-\frac{\tau }{\tau_{s}}+\frac{1}{2\tau_{s}Bi}\right)\mathrm{ch}[(1-\zeta )\sqrt{Bi}]+\frac{\zeta }{2\tau_{s}\sqrt{Bi}}\mathrm{sh}[(1-\zeta )\sqrt{Bi}]\right\}+\\ & +\frac{{\cos\mathrm{ech}}^{2}(\sqrt{Bi})}{2\tau_{s}Bi}\mathrm{ch}(\zeta \sqrt{Bi})-\left(1+\frac{1}{\tau_{s}Bi}\right)\frac{e^{-Bi\;\tau }}{Bi}-2\sum_{n=1}^{\infty }\left(1+\frac{1}{\tau_{s}Bi_{n}}\right)\frac{e^{-Bi_{n}\tau }}{Bi_{n}}\cos(\lambda_{n}\zeta ), \end{array}\;0\leq \zeta \leq 1,\;0\leq \tau \leq \tau_{s}$$

On the heated surface ζ=0 from the solution in Equation (30), it follows that [27]:(31)$$T^{*}(0,\tau )=\frac{\mathrm{cth}(\sqrt{Bi})}{\sqrt{Bi}}\left(1-\frac{\tau }{\tau_{s}}+\frac{1}{2\tau_{s}Bi}\right)+\frac{{\cos\mathrm{ech}}^{2}(\sqrt{Bi})}{2\tau_{s}Bi}-\left(1+\frac{1}{\tau_{s}Bi}\right)\frac{e^{-Bi\;\tau }}{Bi}-2\sum_{n=1}^{\infty }\left(1+\frac{1}{\tau_{s}Bi_{n}}\right)\frac{e^{-Bi_{n}\tau }}{Bi_{n}},\;0\leq \tau \leq \tau_{s}$$
where the coefficients $Bi_{n}$, n=1,2,…. are determined by Equation (27).

In the limiting case $\tau_{s}\to \infty$, the solutions in Equations (30) and (31) take the form:(32)$$T^{*}(\zeta ,\tau )=\frac{\cos\mathrm{ech}(\sqrt{Bi})}{\sqrt{Bi}}\mathrm{ch}[(1-\zeta )\sqrt{Bi}]-\frac{e^{-Bi\;\tau }}{Bi}-2\sum_{n=1}^{\infty }\frac{e^{-Bi_{n}\tau }}{Bi_{n}}\cos(\lambda_{n}\zeta ),\;0\leq \zeta \leq 1,\;\tau \geq 0$$
(33)$$T^{*}(0,\tau )=\frac{\mathrm{cth}(\sqrt{Bi})}{\sqrt{Bi}}-\frac{e^{-Bi\;\tau }}{Bi}-2\sum_{n=1}^{\infty }\frac{e^{-Bi_{n}\tau }}{Bi_{n}},\;\tau \geq 0$$

Next, we verify the achieved solutions. For this purpose, from the solution in Equation (30), we find the derivative:(34)$$\begin{array}{l} \frac{\partial T^{*}(\zeta ,\tau )}{\partial \zeta }=-\frac{\cos\mathrm{ech}(\sqrt{Bi})}{\sqrt{Bi}}\left\{\left(1-\frac{\tau }{\tau_{s}}\right)\sqrt{Bi}\;\mathrm{sh}[(1-\zeta )\sqrt{Bi}]+\frac{\zeta }{2\tau_{s}}\mathrm{ch}[(1-\zeta )\sqrt{Bi}]\right\}+\\ \;\;\;\;\;\;\;\;\;\;\;\;\;\;\;\;\;\;+\frac{{\cos\mathrm{ech}}^{2}(\sqrt{Bi})}{2\tau_{s}\sqrt{Bi}}\mathrm{sh}(\zeta \sqrt{Bi})+2\sum_{n=1}^{\infty }\left(1+\frac{1}{\tau_{s}Bi_{n}}\right)\frac{e^{-Bi_{n}\tau }}{Bi_{n}}\lambda_{n}\sin(\lambda_{n}\zeta ), \end{array}\;0\leq \zeta \leq 1,\;0\leq \tau \leq \tau_{s}$$

On the heated ζ=0 and adiabatic ζ=1 surfaces of the disc, from the relation in Equation (34), it follows that:(35)$${\frac{\partial T^{*}(\zeta ,\tau )}{\partial \zeta }|}_{\zeta =0}=-\left(1-\frac{\tau }{\tau_{s}}\right),\;0\leq \tau \leq \tau_{s}$$
(36)$${\frac{\partial T^{*}(\zeta ,\tau )}{\partial \zeta }|}_{\zeta =1}=-\frac{\cos\mathrm{ech}(\sqrt{Bi})}{2\tau_{s}\sqrt{Bi}}+\frac{{\cos\mathrm{ech}}^{2}(\sqrt{Bi})}{2\tau_{s}\sqrt{Bi}}\mathrm{sh}(\sqrt{Bi})=0,\;0\leq \tau \leq \tau_{s}$$
i.e., the solution in Equation (30) satisfies the boundary conditions in Equations (12) and (13). Substituting τ=0 in Equation (30), we get:(37)$$\begin{array}{l} T^{*}(\zeta ,0)=\frac{\cos\mathrm{ech}(\sqrt{Bi})}{\sqrt{Bi}}\left\{\left(1+\frac{1}{2\tau_{s}Bi}\right)\mathrm{ch}[(1-\zeta )\sqrt{Bi}]+\frac{\zeta }{2\tau_{s}\sqrt{Bi}}\mathrm{sh}[(1-\zeta )\sqrt{Bi}]\right\}+\\ \;\;\;\;\;\;\;\;\;\;\;\;\;\;\;+\frac{{\cos\mathrm{ech}}^{2}(\sqrt{Bi})}{2\tau_{s}Bi}\mathrm{ch}(\zeta \sqrt{Bi})-\left(1+\frac{1}{\tau_{s}Bi}\right)\frac{1}{Bi}-2\sum_{n=1}^{\infty }\left(1+\frac{1}{\tau_{s}Bi_{n}}\right)\frac{\cos(\lambda_{n}\zeta )}{Bi_{n}} \end{array},0\leq \zeta \leq 1$$

Using the sums (28) and (29), we have:(38)$$\begin{array}{ll} 2\sum_{n=1}^{\infty }\left(1+\frac{1}{\tau_{s}Bi_{n}}\right)\frac{\cos(\lambda_{n}\zeta )}{Bi_{n}}= & -\frac{1}{Bi}-\frac{1}{\tau_{s}Bi^{2}}+\frac{{\cos\mathrm{ech}}^{2}(\sqrt{Bi})}{2\tau_{s}Bi}\mathrm{ch}(\zeta \sqrt{Bi})+\\ & +\frac{\cos\mathrm{ech}(\sqrt{Bi})}{\sqrt{Bi}}\left\{\left(1+\frac{1}{2\tau_{s}Bi}\right)\mathrm{ch}[(1-\zeta )\sqrt{Bi}]+\frac{\zeta }{2\tau_{s}\sqrt{Bi}}\mathrm{sh}[(1-\zeta )\sqrt{Bi}]\right\}, \end{array}\;0\leq \zeta \leq 1$$

Introducing the sum in Equation (38) into Equation (37), we confirms the execution of the initial condition in Equation (14).

Finally, taking into account the notation in Equation (10), we obtain the temperature field in the disc:(39)$$T(z,t)=T_{a}+T_{0}T^{*}(\zeta ,\tau ),0\leq z\leq d,0\leq t\leq t_{s}$$
where $T^{*}(\zeta ,\tau )$,0≤ζ≤1,$0\leq \tau \leq \tau_{s}$ is the dimensionless temperature (Equation (30)).

## 4. Thermal Insulation of the Lateral Surfaces of the Disc

Formally, for h=0, the boundary-value problem of heat conduction in Equations (11)–(15), with respect to dimensionless temperature $T^{*}(\zeta ,\tau )$, becomes similar to the problem in Equations (17)–(20) with respect to the function Θ(ζ,τ). Therefore, analogously to Equation (24), we can write:(40)$$T^{*}(\zeta ,\tau )=J_{0}(\tau )+2\sum_{n=1}^{\infty }e^{-\lambda_{n}{}^{2}\tau }J_{n}(\tau )\cos(\lambda_{n}\zeta ),\;0\leq \zeta \leq 1,\;0\leq \tau \leq \tau_{s}$$
where
(41)$$J_{0}(\tau )=\tau \left(1-\frac{\tau }{2\tau_{s}}\right)$$
(42)$$J_{n}(\tau )=-\frac{1}{\lambda_{n}^{2}}\left(1+\frac{1}{\tau_{s}\lambda_{n}^{2}}\right)+\left(1-\frac{\tau }{\tau_{s}}+\frac{1}{\tau_{s}\lambda_{n}^{2}}\right)\frac{e^{\lambda_{n}^{2}\;\tau }}{\lambda_{n}^{2}}$$
and coefficients $\lambda_{n}$, n=1,2,…. have the form in Equation (23).

With account of the sums [35]
(43)$$\sum_{n=1}^{\infty }\frac{\cos(\lambda_{n}\zeta )}{\lambda_{n}^{2}}=\frac{1}{2}\left(\frac{1}{2}\zeta^{2}-\zeta +\frac{1}{3}\right),\;0\leq \zeta \leq 1$$
(44)$$\sum_{n=1}^{\infty }\frac{\cos(\lambda_{n}\zeta )}{\lambda_{n}^{4}}=-\frac{1}{6}\left(\frac{1}{8}\zeta^{4}-\frac{1}{2}\zeta^{3}+\frac{1}{2}\zeta^{2}-\frac{1}{15}\right),\;0\leq \zeta \leq 1$$
the solution in Equations (40)–(42) takes the form:(45)$$\begin{array}{ll} T^{*}(\zeta ,\tau )= & \tau \left(1-\frac{\tau }{2\tau_{s}}\right)+\left(\frac{1}{2}\zeta^{2}-\zeta +\frac{1}{3}\right)\left(1-\frac{\tau }{\tau_{s}}\right)-\\ & \frac{1}{3\tau_{s}}\left(\frac{1}{8}\zeta^{4}-\frac{1}{2}\zeta^{3}+\frac{1}{2}\zeta^{2}-\frac{1}{15}\right)-2\sum_{n=1}^{\infty }\left(1+\frac{1}{\tau_{s}\lambda_{n}^{2}}\right)\frac{e^{-\lambda_{n}^{2}\tau }}{\lambda_{n}^{2}}\cos(\lambda_{n}\zeta ) \end{array},\;0\leq \zeta \leq 1,\;0\leq \tau \leq \tau_{s}$$

Substituting ζ=0 in the solution in Equation (45), we find known formula to calculate the dimensionless temperature of the heated surface of the adiabatic disc [27]:(46)$$T^{*}(0,\tau )=\tau \left(1-\frac{\tau }{2\tau_{s}}\right)+\frac{1}{3}\left(1-\frac{\tau }{\tau_{s}}+\frac{1}{15\tau_{s}}\right)-2\sum_{n=1}^{\infty }\left(1+\frac{1}{\tau_{s}\lambda_{n}^{2}}\right)\frac{e^{-\lambda_{n}^{2}\tau }}{\lambda_{n}^{2}},\;0\leq \tau \leq \tau_{s}$$

## 5. Effective Thermal Conductivity Coefficients

Let us consider the brake disc made from CFCM Thermal-ADF. It consists of the bundles with fibers incorporated into the matrix. Cylindrical shaped bundles are arranged in layers parallel to the front surfaces of the disc. Effective values of thermal conductivities of such composite $K_{z}$ and $K_{r}$ are determined by means of averaging at the three scale levels: micro, meso, and macro [28].

At the micro-level, a single bundle is considered, which consists of closely spaced fibers embedded in the matrix material (Figure 2).

The effective coefficients of thermal conductivity of this bundle in the transversal direction (across the fibers) ${\hat{K}}_{⊥}$ and in longitudinal direction (along the fibers) ${\hat{K}}_{\mathrm{II}}$ are established using the mixture rule in the following form [36]:(47)$${\hat{K}}_{⊥}={\left(\frac{V_{f}}{K_{f}}+\frac{1-V_{f}}{K_{m}}\right)}^{-1},\;{\hat{K}}_{\mathrm{II}}=V_{f}K_{f}+(1-V_{f})K_{m}$$
where $K_{f}$ and $K_{m}$ are the thermal conductivities of fiber material and matrix material, respectively, and $0\leq V_{f}\leq 1$ is the fiber volume ratio.

A unit cell of composite at the meso-level has the shape of a rectangular cuboid with length A and squared cross-section with width B. This cell contains a single centered bundle in form of squared cuboid with dimensions a and b, respectively (Figure 3). It should be noted that the real circular shape of bundle cross-section has been replaced by a square to simplify further calculations, and it does not introduce significant errors in calculations [28].

The effective coefficients of thermal conductivity for selected unit cell of composite in longitudinal $K_{\mathrm{II}}$ and transversal $K_{⊥}$ directions are determined using the method of heat flux lines linearization [37]. This method consists in the division of the unit cell of composite by adiabatic or isothermal and adiabatic planes and calculating the thermal resistance of each part:(48)$$R=\frac{l}{SK}$$
where l is the distance traveled by a heat flux Q, S is the area of cross-section of the part perpendicular to the heat flux direction, and K is the thermal conductivity of the material of the same part (Figure 4). Bearing in mind that the results depend on the manner of the unit cell division, we find effective thermal conductivities of cell using two methods of division by means of adiabatic and isothermal planes. The final results are established as an arithmetic average of two outcomes obtained by different methods. Below, we consider these two methods.

Division of a unit cell by adiabatic planes 1-1 and 2-2 in the transverse direction is shown in Figure 4a, and in the longitudinal direction in Figure 4b.

Based on Equation (48), the effective coefficients of thermal conductivity of selected unit cell of composite in this case are determined in the form:(49)$$K_{⊥}=\frac{B}{ABR_{⊥}}=\frac{1}{AR_{⊥}},\;K_{\mathrm{II}}=\frac{A}{B^{2}R_{\mathrm{II}}}$$
where, according to the scheme of combining resistances $R_{i}$, i=1,2,3 of individual part (Figure 5a), the total thermal resistances $R_{⊥}$ and $R_{\mathrm{II}}$ of the unit cell are equal to:(50)$$R_{⊥}=\frac{(2R_{⊥,1}+R_{⊥,2})R_{⊥,3}}{2R_{⊥,1}+R_{⊥,2}+R_{⊥,3}}={\left(\frac{1}{R_{⊥,3}}+\frac{1}{2R_{⊥,1}+R_{⊥,2}}\right)}^{-1}$$
(51)$$R_{⊥,1}=\frac{B-b}{2abK_{m}},\;R_{⊥,2}=\frac{1}{a{\hat{K}}_{⊥}},\;R_{⊥,3}=\frac{B}{(AB-ab)K_{m}}$$
(52)$$R_{\mathrm{II}}=\frac{(2R_{\mathrm{II},1}+R_{\mathrm{II},2})R_{\mathrm{II},3}}{2R_{\mathrm{II},1}+R_{\mathrm{II},2}+R_{\mathrm{II},3}}={\left(\frac{1}{R_{\mathrm{II},3}}+\frac{1}{2R_{\mathrm{II},1}+R_{\mathrm{II},2}}\right)}^{-1}$$
(53)$$R_{\mathrm{II},1}=\frac{A-a}{2b^{2}K_{m}},\;R_{\mathrm{II},2}=\frac{a}{b^{2}{\hat{K}}_{\mathrm{II}}},\;R_{\mathrm{II},3}=\frac{A}{(B^{2}-b^{2})K_{m}}$$
and the thermal conductivities ${\hat{K}}_{\mathrm{II}}$ and ${\hat{K}}_{⊥}$ of the bundle contained in the unit cell are calculated using Equation (47).

Substituting the effective thermal resistance $R_{⊥}$ (Equations (50) and (51)) and $R_{\mathrm{II}}$ (Equations (53) and (540) into Equation (49), we find:(54)$$K_{⊥}=\left(1-\frac{ab}{AB}\right)K_{m}+\frac{1}{A}{\left(\frac{B-b}{abK_{m}}+\frac{1}{a{\hat{K}}_{⊥}}\right)}^{-1},$$
(55)$$K_{\mathrm{II}}=\left(1-\frac{b^{2}}{B^{2}}\right)K_{m}+\frac{A}{B^{2}}{\left(\frac{A-a}{b^{2}K_{m}}+\frac{a}{b^{2}{\hat{K}}_{\mathrm{II}}}\right)}^{-1}.$$

Division of the unit cell simultaneous by adiabatic 1-1 and 2-2 and isothermal 3-3 and 4-4 planes, respectively, in transverse and longitudinal directions, are presented in Figure 6. A scheme of combining resistances for this division is shown in Figure 5b.

Proceeding as above, we obtain:(56)$$R_{⊥}=2R_{⊥,4}+\frac{R_{⊥,2}R_{⊥,5}}{R_{⊥,2}+R_{⊥,5}}=2R_{⊥,4}+{\left(\frac{1}{R_{⊥,2}}+\frac{1}{R_{⊥,5}}\right)}^{-1}$$
(57)$$R_{⊥,4}=\frac{B-b}{2ABK_{m}},\;R_{⊥,5}=\frac{b}{(AB-ab)K_{m}}$$
(58)$$R_{\mathrm{II}}=2R_{\mathrm{II},4}+\frac{R_{\mathrm{II},2}R_{\mathrm{II},5}}{R_{\mathrm{II},2}+R_{\mathrm{II},5}}=2R_{\mathrm{II},4}+{\left(\frac{1}{R_{\mathrm{II},2}}+\frac{1}{R_{\mathrm{II},5}}\right)}^{-1}$$
(59)$$R_{\mathrm{II},4}=\frac{A-a}{2B^{2}K_{m}},\;R_{\mathrm{II},5}=\frac{a}{(B^{2}-b^{2})K_{m}}$$
where thermal resistances $R_{⊥,2}$ and $R_{\mathrm{II},2}$ are determined using Equations (51) and (53). Taking into account the effective thermal resistances $R_{⊥}$ (Equations (56) and (57)) and $R_{\mathrm{II}}$ (Equations (58) and (59)) in Equation (49), we find:(60)$$K_{⊥}=\frac{1}{A}{\left[\frac{B-b}{ABK_{m}}+{\left(a{\hat{K}}_{⊥}+\frac{\left(AB-ab\right)}{b}K_{m}\right)}^{-1}\right]}^{-1}$$
(61)$$K_{\mathrm{II}}=\frac{A}{B^{2}}{\left[\frac{A-a}{B^{2}K_{m}}+{\left(\frac{b^{2}}{a}{\hat{K}}_{\mathrm{II}}+\frac{\left(B^{2}-b^{2}\right)}{a}K_{m}\right)}^{-1}\right]}^{-1}$$

The length A and the width B of the unit cell of composite are related with dimensions a and b of the single bundle and the bundle volume ratio $0\leq V_{b}\leq 1$ as:(62)$$a-b=A-B,\;ab^{2}=V_{b}AB^{2}$$

Eliminating A in Equation (62), we obtain the following algebraic equation for parameter B:(63)$$B^{3}+(a-b)B^{2}-ab^{2}V_{b}^{-1}=0$$

At the macro-level, the fiber bundles are arranged on the planes parallel to the front surfaces of the disc (Figure 7).

The thermal conductivity of the disc in the axial direction is equal to the effective coefficient of thermal conductivity of the unit composite cell in the transverse direction on the meso-level:(64)$$K_{z}=K_{⊥}$$
where $K_{⊥}$ is a mean arithmetic of values calculated from Equations (54) and (60).

When determining the effective coefficient of thermal conductivity of the disc in the radial direction, the orientation of fiber bundles on the planes should be taken into account (Figure 8).

If the bundles are arranged in the radial direction (Figure 8a), then:(65)$$K_{r}=K_{\mathrm{II}}$$
where $K_{\mathrm{II}}$ is a mean arithmetic values obtained from Equations (55) and (61). However, if the bundles are oriented in circumferential direction (Figure 8b), then:(66)$$K_{r}=K_{⊥}$$
where $K_{⊥}$ is determined as in Equation (64). In the case of composite discs with chaotically oriented fiber bundles (Figure 8c), the effective coefficient of thermal conductivity in the radial direction is defined as:(67)$$K_{r}=0.5(K_{⊥}+K_{\mathrm{II}})$$
where $K_{\mathrm{II}}$ and $K_{⊥}$ are calculated in the same manner as in Equations (65) and (66), respectively.

The empirical Equation (67) assumes that the number of bundles oriented in the radial and circumferential directions are equal. Therefore, the value of $K_{r}$ found using Equation (67) for the chaotic orientation bundles may be established with an error. One of the proposals to modify Equation (67) could be to use statistical methods with the known probability distribution of bundles orientation.

## 6. Numerical Analysis

Calculations of the temperature T (Equation (39)) were performed for a disc made of carbon composite material^−^ Termar-ADF with randomly oriented fiber bundles ($K_{f}=250\;W\;m^{-1}K^{-1}$, $K_{m}=10\;W\;m^{-1}K^{-1}$, $c=1400\;J\;{\mathrm{kg}}^{-1}K^{-1}$, and $\rho =1800\;\mathrm{kg}\;m^{-3}$) [27]. The values of others input parameters were equal to: d=14 mm, $r_{1}=27\;\mathrm{mm}$, $r_{2}=37\;\mathrm{mm}$, f=0.28, $p_{0}=0.98\;\mathrm{MPa}$, $\omega_{0}=593.75\;\mathrm{rad}\;s^{-1}$, $t_{s}=6.8\;s$, b=1 mm, $V_{f}=0.95$, and $T_{a}=20\;{}^{\circ}C$ [28].

The change of temperature during braking on the friction surface of the disc is presented in Figure 9. Solid line demonstrates the results obtained using Equations (31) and (39) with effective values of thermal conductivities $K_{z}$ (Equation (64)) and $K_{r}$ (Equation (67)). Dotted line presents appropriate data obtained from measurement of temperature by means of thermocouples [27]. It can be seen that the curves are close to each other. In the initial stage of braking, the temperature achieved in the theoretical solution rapidly increases to maximum value $T_{\max}=466{\;}^{\circ }C$ at $t=t_{\max}=3.36\;s$, whereupon it decreases to $T=337{\;}^{\circ }C$ at the stop $t=t_{s}=6.8\;s$. The highest value of the experimentally measured temperature is almost the same; however, it is attained a bit later at $t=t_{\max}\approx 4\;s$. In the final stage of braking, the experimental temperature is slightly higher than the value obtained using the theoretical solution.

Changes of temperature during braking on different distances from heated surface of the disc are shown in Figure 10. Calculations were conducted based on the exact solution in Equations (30) and (39). The temperature growth at the very beginning of the process is the fastest, and its maximum value is the highest on the friction surface z=0. Increasing the distance from this surface to the inside of the disc, the temperature decreases and the delay effect occurs—elongation of the time $t_{\max}$ of maximum temperature achievement. After achieving the highest temperature value at the particular depths in the range 0≤z<6 mm, the temperature drop lasts until standstill, whereas, for z≥6 mm, the temperature monotonically rises during the whole braking process.

The increase of heat exchange with an environment leads to linear decrease of maximum temperature $T_{\max}$ of the friction surface of the disc (Figure 11). The highest value $T_{\max}=473{\;}^{\circ }C$ is achieved at $t_{\max}=3.39\;s$ for $h=50\;{\mathrm{Wm}}^{-2}K^{-1}$. Enhancement of convective cooling intensity to $h=250\;{\mathrm{Wm}}^{-2}K^{-1}$ results in the drop of the maximum temperature to $T_{\max}=460{\;}^{\circ }C$ at $t_{\max}=3.29\;s$.

Changes of effective thermophysical properties of composite material due to increasing the length a of bundles are shown in Figure 12. We see that significant rise of $K_{r}$ and drop of $K_{z}$ and k take place in the range 1 mm≤a≤15 mm. Further increase of parameter a causes gentler variations, and for a≥25 mm values of coefficients $K_{r}$, $K_{z}$, and k remain almost constant. Since the bundles are located on the planes oriented parallel to the front disc surface z=0 (Figure 7), their length has slight effect on heat conduction in the axial direction. For the shortest considered length a=1 mm, the effective coefficient of thermal conductivity in this direction has the greatest value $K_{z}=28.3\;W\;m^{-1}K^{-1}$. Inconsiderable elongation of bundles results in the slight decrease to $K_{z}\approx 25\;W\;m^{-1}K^{-1}$, and it remains steady on this level within 7 mm≤a≤30 mm. The composite’s ability to conduct heat in the radial direction $K_{r}$ significantly increases from $K_{r}=28.7\;W\;m^{-1}K^{-1}$ for a=1 mm to $K_{r}=63.5\;W\;m^{-1}K^{-1}$ for a=30 mm. On the contrary to the described above changes of effective thermal conductivity, the thermal diffusivity is the highest $k=1.125\times 10^{-5}\;m^{2}s^{-1}$ for the composite with the shortest bundles (a=1 mm). With increase of the bundle length, effective thermal diffusivity drops, attaining minimum value $k=0.985\times 10^{-5}\;m^{2}s^{-1}$ for a=30 mm.

With increasing bundle length, the maximum temperature of the heated disc surface z=0 is rising from the value $T_{\max}=439.4{\;}^{\circ }C$ for a=1 mm to $T_{\max}=459.2{\;}^{\circ }C$ for a=30 mm(Figure 13). As shown in Figure 12, the coefficients of thermal conductivity in radial and axial directions of composite with short fiber bundles take close values and its thermal diffusivity is the greatest. The last leads to a faster heat removal from the surface of friction and, consequently, to relatively low value of maximum temperature. Increase of fiber bundles causes decrease of thermal diffusivity and increase of maximum temperature. Growth of $T_{\max}$ is the most rapid over the change range 1<a≤5 mm, when the drop of thermal diffusivity is the largest.

Increase of concentration of bundles in composite results in increase of its thermal diffusivity and conductivity from values $k=0.985\times 10^{-5}\;m^{2}s^{-1}$, $K_{r}=63.5\;{\mathrm{Wm}}^{-1}K^{-1}$, and $K_{z}=25\;{\mathrm{Wm}}^{-1}K^{-1}$ at $V_{p}=0.5$ to $k=3.527\times 10^{-5}\;m^{2}s^{-1}$, $K_{r}=155.9\;{\mathrm{Wm}}^{-1}K^{-1}$, and $K_{z}=88.9\;{\mathrm{Wm}}^{-1}K^{-1}$ at $V_{b}=0.95$ (Figure 14). Such growth of material ability to dissipate heat from friction surface of the disc causes almost linear drop of maximum temperature value $T_{\max}$ and elongation of time $t_{\max}$ of its achievement (Figure 15). For the smallest considered concentration of fiber bundles in composite ($V_{b}=0.5$), the maximum temperature $T_{\max}=466{\;}^{\circ }C$ is the highest and attained at time $t_{\max}=3.36\;s$, while, for the largest value $V_{b}=0.95$, we have $T_{\max}=289.2{\;}^{\circ }C$ and $t_{\max}=4.73\;s$.

## 7. Conclusions

The calculation scheme is proposed to estimate the maximum temperature of a multi-disc brake system. For this purpose, the boundary-value problem of heat conduction is formulated for a single disc considering the frictional heating on its front surfaces, convective cooling of its lateral surfaces, and the detailed structure of the composite material. This material is made up of layers of fiber bundles arranged in planes parallel to the front surfaces of the disc. Each bundle has a cylindrical shape filled with matrix material reinforced with carbon fibers. The effective thermal conductivities of this composite are calculated at the three scale levels: micro, meso, and macro. The exact solution of the problem is obtained, which allows investigating variations of disc temperature under the effect of input parameters of braking process and friction material parameters (such as the concentration of fibers and their dimensions and orientation). Calculations were carried out for a disc made up of carbon composite material Termar-ADF, widely applied in aircraft brakes. Due to performed numerical analysis, it is established that:Temperature values calculated on the basis of the obtained analytical solution are consistent with appropriate experimental data presented in [27].During short-lasting ($0<t_{s}\leq 10\;s$), heavy loaded ($p_{0}≅1\;\mathrm{MPa}$) braking processes, the influence of convective cooling of the lateral disc surfaces on its maximum temperature is inconsiderable. The decrease of the maximum temperature value with increase of coefficient of heat exchange is linear.The increase of the concentration of fiber bundles in the composite results in increase of its thermal conductivity and diffusivity.The high temperature of the disc at fixed intensity of convective cooling can be decreased by using shorter fiber bundles and simultaneously increasing their concentration in the composite material.

The solution was obtained without considering possible variations of friction coefficient and material properties under the influence of temperature. It should be noted that consideration of thermal sensitivity and friction coefficient changes is possible by using the nonlinear models, which require numerical methods, such as FEM, to solve the proper thermal problems of friction. Some steps are taken to do this in [29], although for a different material. Another problem that is difficult to solve, arising in this context, is related to the development of composite structure models with temperature-dependent thermophysical properties of its components. Even in the considered case of constant thermophysical properties, the procedure for determining the effective properties of the composite is quite complicated. We achieved this by using some geometric simplifications, in particular by converting the actual circular cross-sectional shape of the bundle into a square. The development of mathematical models of the composite, taking into account thermal sensitivity of its components and their interaction will be one of the directions of our future research.

Based on these findings, we conclude that the proposed mathematical model allows with sufficient accuracy determining a temperature mode of a multi-disc brake at a single braking. Calculations are executed only for one frictional composite material, but the model proposed can be used for other materials, too. In such a case, it may be necessary to use other methods for obtaining effective thermophysical properties of the composite [38].

## Figures and Tables

**Figure 1 materials-13-02691-f001:**
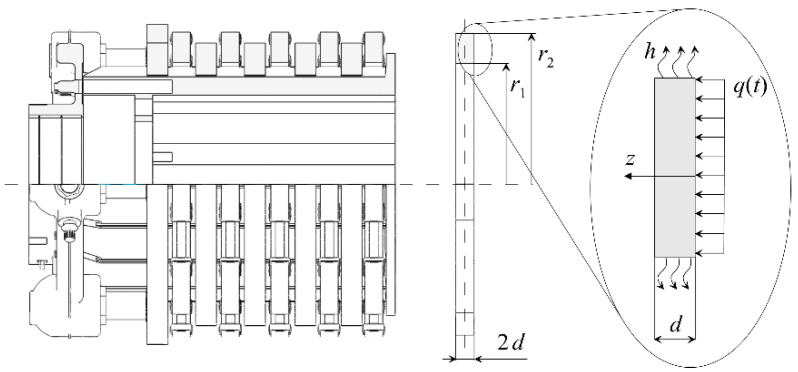
Disc brake: scheme of the problem.

**Figure 2 materials-13-02691-f002:**
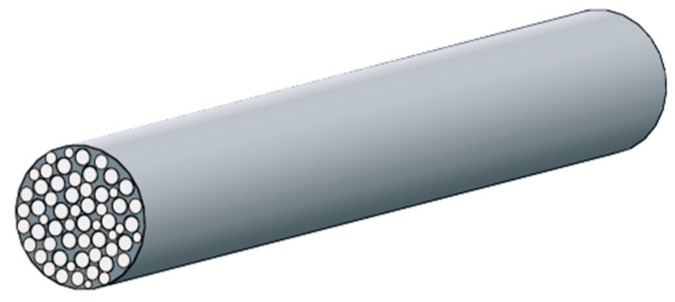
Single bundle with fibers (the micro-level).

**Figure 3 materials-13-02691-f003:**
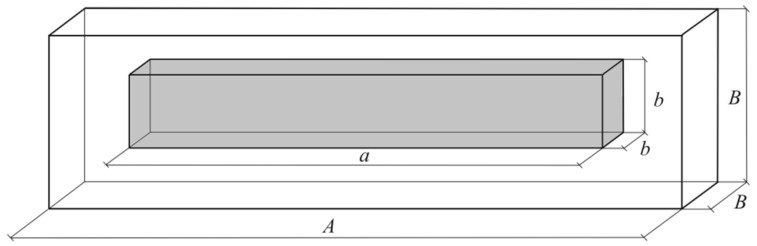
Unit cell of composite (the meso-level).

**Figure 4 materials-13-02691-f004:**
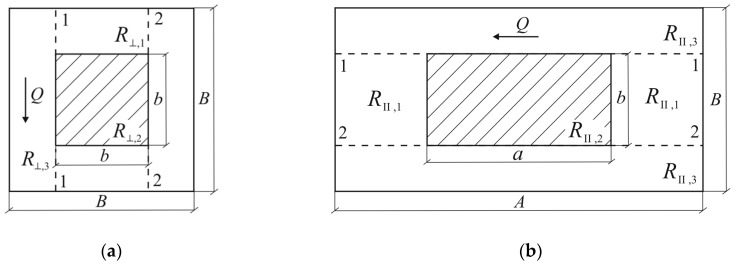
Division of a unit cell of composite by adiabatic planes 1-1, 2-2 in: (**a**) transverse direction; and (**b**) longitudinal direction. The arrows indicate the direction of heat flux Q.

**Figure 5 materials-13-02691-f005:**
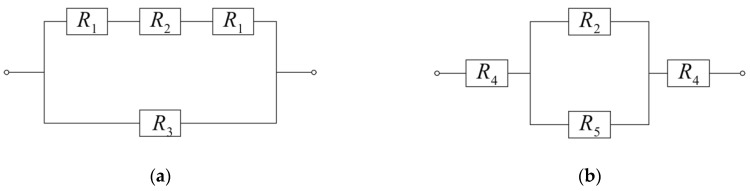
Scheme of combining the particular resistances for the divisions of a unit cell by planes: (**a**) adiabatic; and (**b**) adiabatic and isothermal.

**Figure 6 materials-13-02691-f006:**
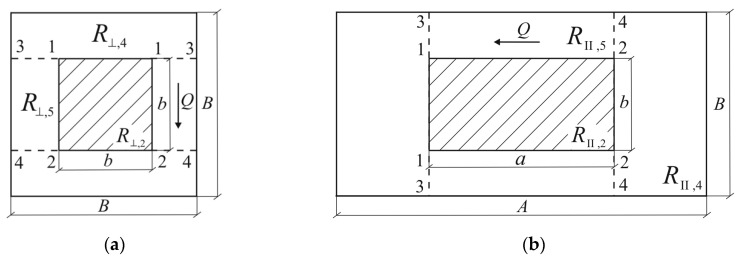
Division of the unit cell of composite by adiabatic 1-1, 2-2 and isothermal 3-3, 4-4 planes in: (**a**) transverse direction; and (**b**) longitudinal direction. The arrows indicate the direction of heat flux Q.

**Figure 7 materials-13-02691-f007:**
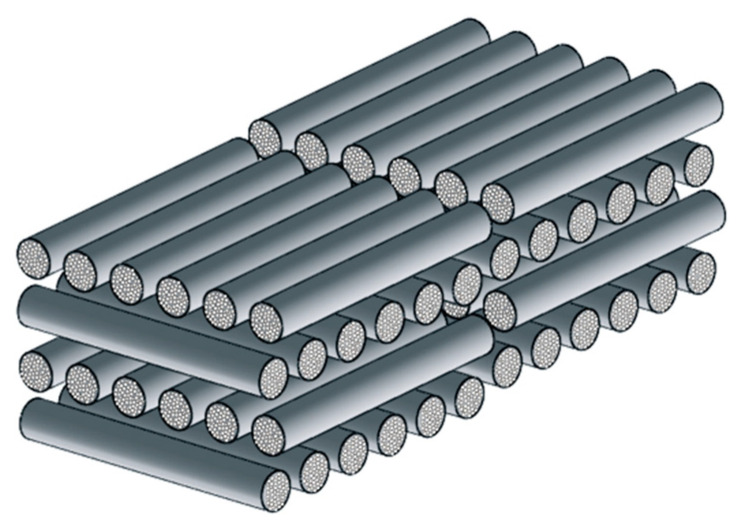
Layers of the bundles (the macro-level).

**Figure 8 materials-13-02691-f008:**
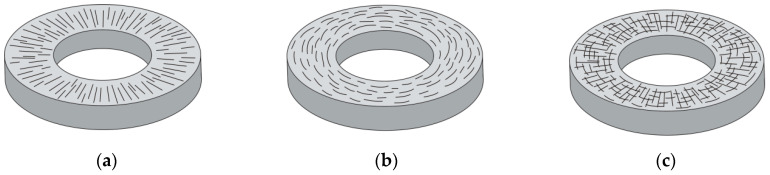
Orientation of bundles in the disc: (**a**) radial; (**b**) circumferential; and (**c**) random.

**Figure 9 materials-13-02691-f009:**
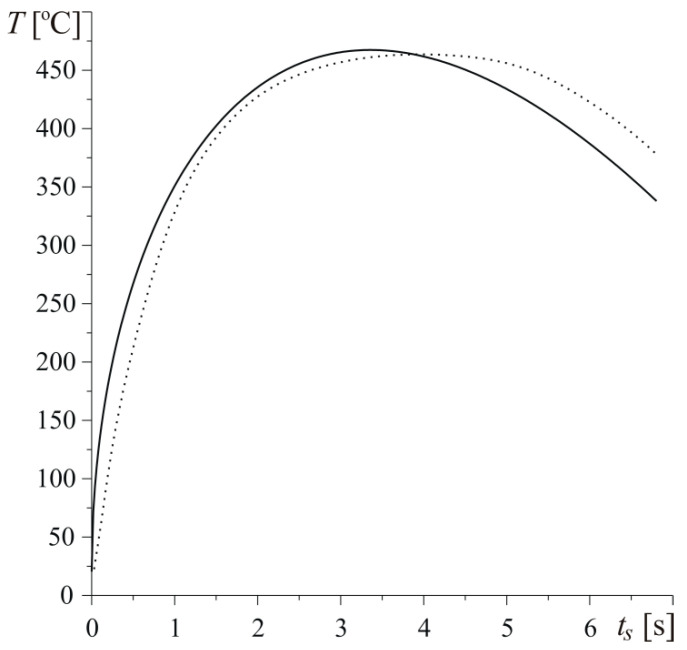
Evolutions of temperature T on the heated surface z=0 of the disc for $h=140\;W\;m^{-2}K^{-1}$, a=30 mm, and $V_{b}=0.5$. The solid line is the theoretical solution using Equations (31) and (39) and the dotted line is the experimental data from the article [27].

**Figure 10 materials-13-02691-f010:**
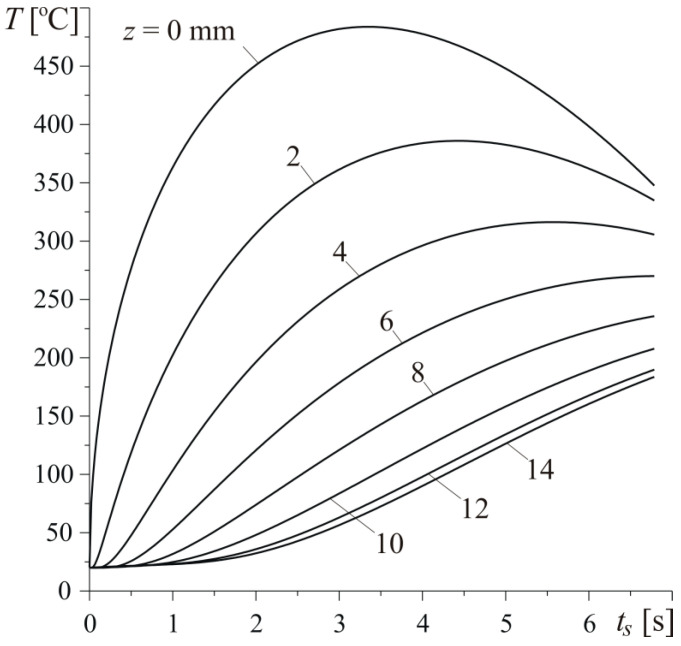
Evolutions of temperature T on the different depths z from the heated surface of the disc for $h=140\;W\;m^{-2}K^{-1}$, a=30 mm, and $V_{b}=0.5$.

**Figure 11 materials-13-02691-f011:**
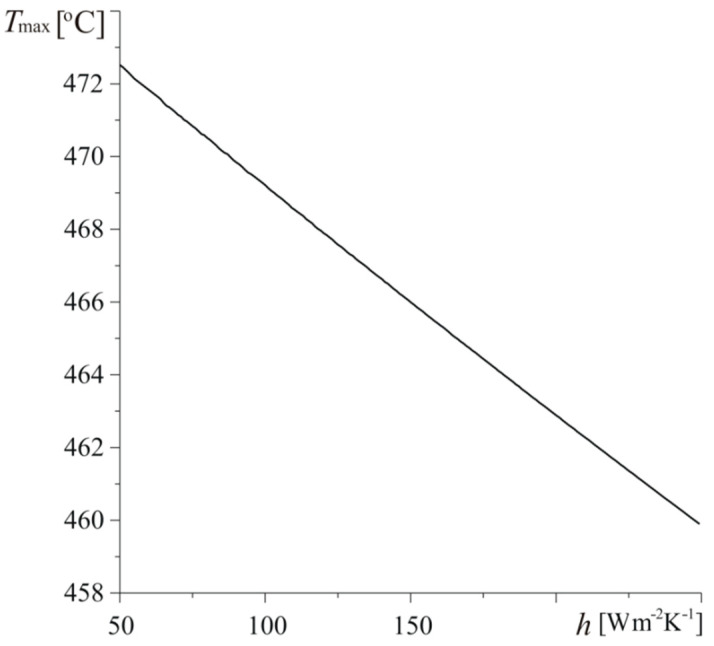
Dependence of maximum temperature $T_{\max}$ of the heated surface of the disc z=0 on coefficient of heat exchange h for a=30 mm,
$V_{b}=0.5$.

**Figure 12 materials-13-02691-f012:**
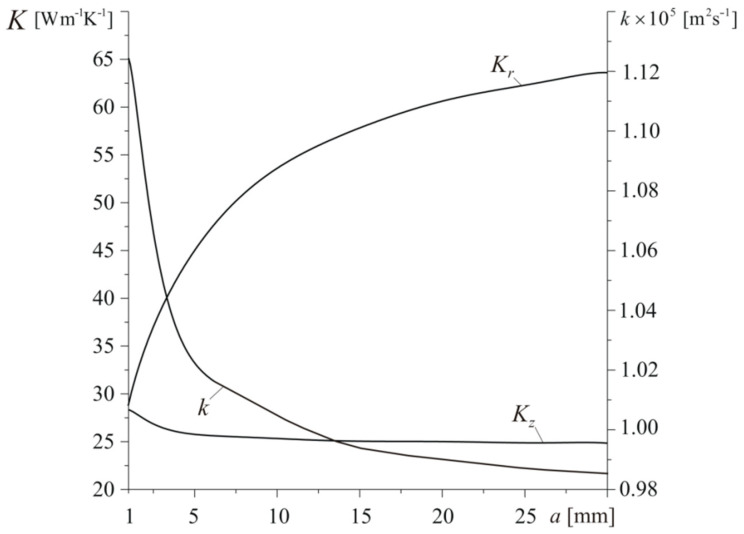
Dependencies of effective coefficients of thermal conductivity $K_{r}$, $K_{z}$ and thermal diffusivity k on the length a of fiber bundle for $h=140\;W\;m^{-2}K^{-1}$ and $V_{b}=0.5$.

**Figure 13 materials-13-02691-f013:**
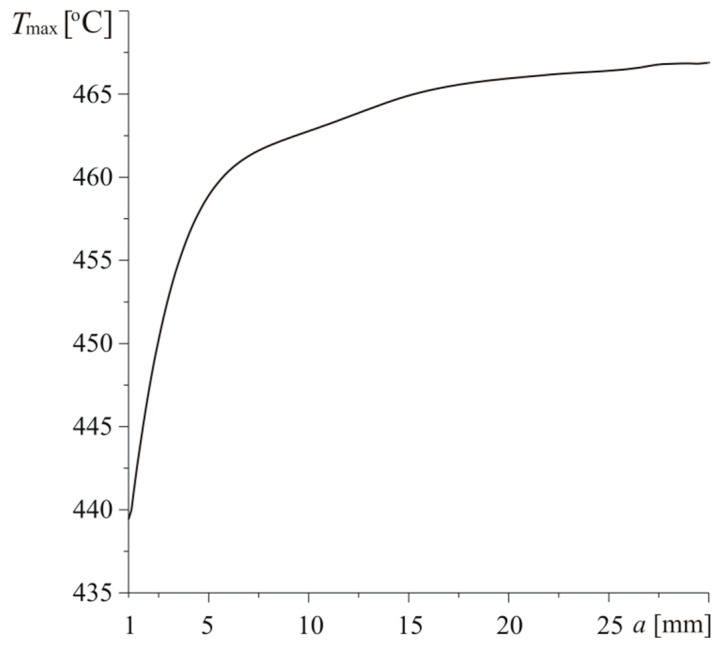
Dependence of maximum temperature $T_{\max}$ of the disc on the length of bundles a for $h=140\;W\;m^{-2}K^{-1}$ and $V_{b}=0.5$.

**Figure 14 materials-13-02691-f014:**
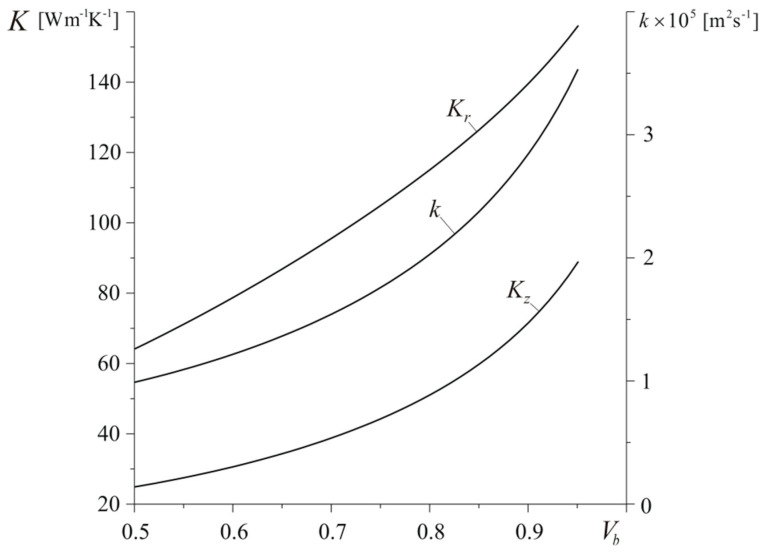
Dependencies of effective thermal conductivities $K_{r}$, $K_{z}$ and thermal diffusivity k on the coefficient of volumetric concentration of bundles $V_{b}$ for $h=140\;W\;m^{-2}K^{-1}$ and a=30 mm.

**Figure 15 materials-13-02691-f015:**
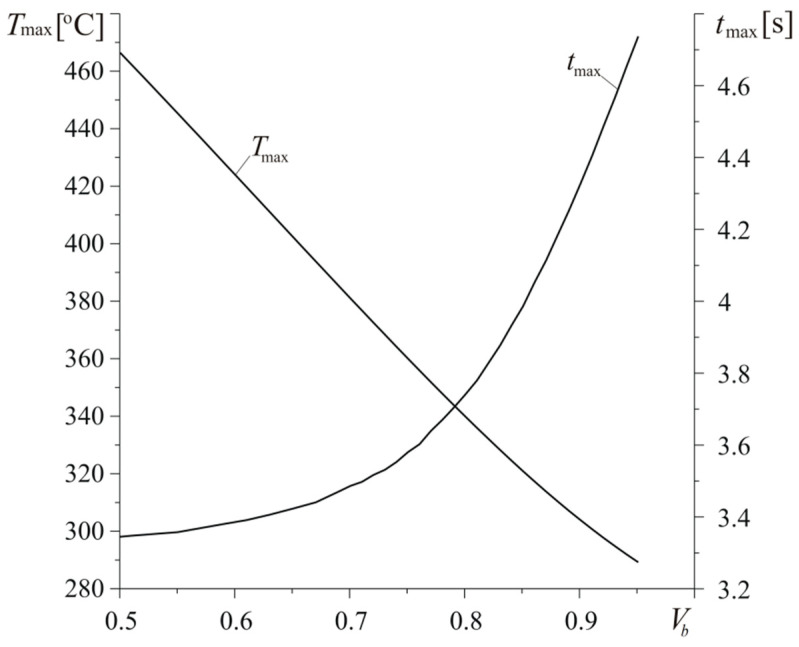
Dependencies of maximum temperature $T_{\max}$ and time of its achievement $t_{\max}$ on the coefficient of volumetric concentration of fiber bundles $V_{b}$ for $h=140\;W\;m^{-2}K^{-1}$ and a=30 mm.

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
