# Peer review of "Frictional Heating during Braking of the C/C Composite Disc"

_materials, 2020, doi:10.3390/ma13122691_

Round 1
Reviewer 1 Report
The analyzed topic is up to date and interesting. The issue is processed on a good level. There are often mentioned data from the literature. It may be useful to complete the text with the authors own practical experience. There are evaluated a lot of expressions. In the expressions and equations, on my opinion, are missing the units of the variables. In text are mentioned the results from the FEM computation. Are the results from the computation relevant as an input to other computation? Did the authors analyze and compare results from their computation, simulation with any results from their own measurement or experimental investigation? For which purpose has the investigation the main benefit. For what real application? If any exist.
Author Response
We hereby would like to thank the reviewer for a favorable reviews and for providing constructive comments. A listed responses to comments are enclosed below. In the revised version of the manuscript answers to remarks of reviewers are noted by blue color.
Reviewer 1
The analyzed topic is up to date and interesting. The issue is processed on a good level. There are often mentioned data from the literature. It may be useful to complete the text with the authors own practical experience. There are evaluated a lot of expressions. In the expressions and equations, on my opinion, are missing the units of the variables. In text are mentioned the results from the FEM computation. Are the results from the computation relevant as an input to other computation? Did the authors analyze and compare results from their computation, simulation with any results from their own measurement or experimental investigation? For which purpose has the investigation the main benefit. For what real application? If any exist.
Comment 1
In the expressions and equations, on my opinion, are missing the units of the variables.
Answer 1
Units of input parameters as well as calculated quantities are given in chapter 6. Taking into account reviewer's comment, the Nomenclature chapter has been added, which includes the units of the variables.
Comment 2
In text are mentioned the results from the FEM computation. Are the results from the computation relevant as an input to other computation?
Answer 2
Numerical calculations using FEM in the article [30] have been made for another material - CFCM-5. In this article we considered Termar-ADF material. We did not carry out numerical simulations using FEM for this material. Our goal was to obtain an exact analytical solution to the problem, which in the future can be used as a benchmark for obtaining relevant results (approximate!) using FEM. In this sense, the results obtained in this article are significant and can be used to verify the results obtained using FEM.
Comment 3
Did the authors analyze and compare results from their computation, simulation with any results from their own measurement or experimental investigation? For which purpose has the investigation the main benefit. For what real application? If any exist.
Answer 3
The composite material considered in this article is not widely available on the commercial market. For this reason, we did not have the opportunity to conduct appropriate experimental research, but only used the results known in the scientific literature, especially from the article [28]. The main advantage of our solution is the ability to quickly estimate the entire temperature field (not only surface temperature) in the disc, taking into account the composite structure of the material.
Reviewer 2 Report
I am recommending the rejection of this manuscript primarily because it makes no argument for its own purpose. The literature review in the introduction makes mention of numerous existing models but fails to motivate the need for one more. Sections 2 and 3 of the manuscript present already existing solutions (ref. [28] for example). Section 5 and 6 present similar approach and results as ref [29].
Author Response
Reviewer 2
I am recommending the rejection of this manuscript primarily because it makes no argument for its own purpose. The literature review in the introduction makes mention of numerous existing models but fails to motivate the need for one more. Sections 2 and 3 of the manuscript present already existing solutions (ref. [28] for example). Section 5 and 6 present similar approach and results as ref [29].
Comment 1
The literature review in the introduction makes mention of numerous existing models but fails to motivate the need for one more.
Answer 1
The justification for the development of analytical models is that the exact solutions obtained on their basis are used to verify the results obtained by approximate methods, including FEM. The analytical model contains some simplifications of the real physical model, but this allows to obtain an exact solution. Simulation models using FEM or other methods, give a better approximation to the physical model, but received solutions are approximate and require, in cases of specific parameters, verification of results.
Comment 2
Sections 2 and 3 of the manuscript present already existing solutions (ref. [28] for example). Section 5 and 6 present similar approach and results as ref [29].
Answer 2
In the introduction, we clearly highlighted the main differences between known analytical solutions, including in the article [28]. This difference consists in obtaining an exact solution for the temperature field in the whole element, and not only for surface temperature. Of course, in the limiting case we receive the results from the article [28], which is appropriately marked in the manuscript. The methodology for determining the effective characteristics of the composite is similar to that presented in the article [29]. The new element of this method is taking into account the two directions (transversal and longitudinal) of division of a unit cell and calculation of effective composite properties as the arithmetic average received from both approaches. In monograph [38] it has been proved that this increases the accuracy of the solution obtained. However in the article [29], only one, transversal direction was taken into account to establish the effective characteristics. In addition, only the final results of calculations are given in the article [29], while we presented in detail each of the two methods used. Presented in the chapter 6 results of numerical calculations are new and do not coincide with the corresponding results from the article [29]. Also it should be noted that in the article [29], the exact solution of the problem is not obtained, merely the general solution is presented in the form of Duhamel’s integral, which can be computed only numerically. It has been written in the introduction.
Reviewer 3 Report
The paper deals with analytical solutions for the heat transport in multi-disc brake systems, as they appear in heavy-duty vehicles and airplanes. The special focus is on the unisotropic and multiscale compution of the used composite material. The paper is well written but I have some basic remarks and suggestions that are listed in the following:
1) I assume that the majority of the readers of "Materials" is not familiar with two-disc brakes as they are discussed in this paper. A drawing that shows this construction would be helpfull. Figure 1, on the other hand, does not provide much new information, as the fibre structure is not really visible either. Are there better microscope images available for this?
2) The derivation of the solution for the equations for a ring-ring contact is documented in great detail. Therein the authors use quotations especially [28] and [36]. If formula (46) is already available in the literature, I think it makes no sense to present this derivation in such detail. Instead, the initial equations, the main assumptions and the final equations with the literature references would be sufficient.
3) I would expect all material parameters to be strongly dependent on temperature, especially at the high temperatures that are reached here. This concerns the thermal conductivity, the heat capacity and also very strongly the coefficient of friction. I would assume that these dependencies are also known (at least by the manufacturers). The authors should include this in their studies if possible.
4) In my opinion, the assumptions behind equation (66) need to be worked out more clearly. I suppose that in reality the arrangement of the fibres is most likely to look like Figure 8c (as also used by the authors in chapter 6). Against this background, additional studies are needed to support this assumption.
5) I understand that the authors approach the question from a very analytical point of view and I appreciate this very much, since nowadays numerical methods are often used unnecessarily. Nevertheless, I could imagine that there are non-linear coupling effects between the fibers, which are consequently not taken into account. The validation of the derived equations is only done by one integral measurement of the temperature. Here I would also like to see comparative calculations with FEM, which in particular confirm the validity of my comment under 4).
6) The conclusions and particularly the outlook are too short for me. The possibilities and limitations are hardly ever reflected, nor is it shown how these results can be validated in the future.
Author Response
We hereby would like to thank the reviewer for a favorable reviews and for providing constructive comments. A listed responses to comments are enclosed below. In the revised version of the manuscript answers to remarks of reviewers are noted by blue color.
Reviewer 3
The paper deals with analytical solutions for the heat transport in multi-disc brake systems, as they appear in heavy-duty vehicles and airplanes. The special focus is on the unisotropic and multiscale compution of the used composite material. The paper is well written but I have some basic remarks and suggestions that are listed in the following:
Comment 1
I assume that the majority of the readers of "Materials" is not familiar with two-disc brakes as they are discussed in this paper. A drawing that shows this construction would be helpfull. Figure 1, on the other hand, does not provide much new information, as the fibre structure is not really visible either. Are there better microscope images available for this?
Answer 1
In the new version of the article we took into account the Reviewer's attention. Unfortunately, we do not have this material, we only use data known in the literature.
Comment 2
The derivation of the solution for the equations for a ring-ring contact is documented in great detail. Therein the authors use quotations especially [28] and [36]. If formula (46) is already available in the literature, I think it makes no sense to present this derivation in such detail. Instead, the initial equations, the main assumptions and the final equations with the literature references would be sufficient.
Answer 2
In the manuscript, we have received an exact solution to the problem, allowing to determine the temperature not only on the friction surface, but also inside the disc. In the limiting case of our solution, we have obtained known from article [28] the formulas (31) and (46) to determine the surface temperature of the disc. We use this to verify our more universal solution. Relevant references to article [28] are given in the manuscript.
Comment 3
I would expect all material parameters to be strongly dependent on temperature, especially at the high temperatures that are reached here. This concerns the thermal conductivity, the heat capacity and also very strongly the coefficient of friction. I would assume that these dependencies are also known (at least by the manufacturers). The authors should include this in their studies if possible.
Answer 3
We completely agree with the reviewer. It should be noted that taking into account the thermal sensitivity of the material and the coefficient of friction is possible under non-linear models, and thus the use of numerical methods, especially FEM or other, to solve relevant frictional heat conduction problems. Some steps in this direction were made in article [30], admittedly, for other material. Another problem arising here is the development of composite structure models with temperature-dependent thermophysical properties of its components. We've added our response to this remark to Summary as one of our future research directions.
Comment 4
In my opinion, the assumptions behind equation (66) need to be worked out more clearly. I suppose that in reality the arrangement of the fibres is most likely to look like Figure 8c (as also used by the authors in chapter 6). Against this background, additional studies are needed to support this assumption.
Answer 4
We agree with this remark. We chose the most simple variant of averaging data obtained using both methods to determine the effective properties of the composite. It should be noted that even in this case we received results quite well in agreement with experimental data (Fig. 9). Thank you to the Reviewer for pointing out one more direction for our research in the future.
Comment 5
I understand that the authors approach the question from a very analytical point of view and I appreciate this very much, since nowadays numerical methods are often used unnecessarily. Nevertheless, I could imagine that there are non-linear coupling effects between the fibers, which are consequently not taken into account. The validation of the derived equations is only done by one integral measurement of the temperature. Here I would also like to see comparative calculations with FEM, which in particular confirm the validity of my comment under 4).
Answer 5
We hope that we partially answered the reviewer's question in p. 4. We would like to emphasize that our goal was to obtain an accurate solution to the complex boundary-value problem, including frictional heating, convective cooling and the structure of the composite material. We achieved this solution for certain simplification assumptions. We compared our results, obtained on the basis of an exact solution, with experimental results and received satisfactory compliance. The question about how much better is the convergence of experimental results with the results obtained using non-linear FEM models will remain open and goes beyond the scope of our study. We take this note as an indication of the next direction of our research in the future - the development of numerical models taking into account the non-linear effects of the interaction between composite fibers.
Comment 6
The conclusions and particularly the outlook are too short for me. The possibilities and limitations are hardly ever reflected, nor is it shown how these results can be validated in the future.
Answer 6
The new version of the manuscript includes this note.
Round 2
Reviewer 2 Report
-
Author Response
Thank you to the reviewer for the opinion about our article.
Reviewer 3 Report
The authors have integrated most of my comments into the outlook, which I can understand and which is certainly okay, but I still miss a more critical handling of the own model in the text. This concerns in particular the assumption from equation (67) as well as the assumption of linearity. I would expect already in the corresponding sections and not only in the outlook that further effects can occur, which have not (yet) been considered. The same applies to the conclusion of Figure 9, where a comparison of the surface temperatures between measurement and simulation is presented. Also this comparison is not self-critical enough for me. Can (with the help of only one measurement, that only provides information on the surface) really conclusions towards the unisotropic material behaviour be drawn? Furthermore, it would be interesting to integrate in this diagram another curve for the calculated case of a homogenized and isotropic material.
Author Response
We hereby would like to thank the reviewer for a favorable review. In the revised version of the manuscript new answers to remarks of reviewer are noted by green color.
In the new version of the manuscript, we have included the reviewer's remark regarding the assumption about the linearity of the adopted model and the resulting possibilities of using the solutions obtained (chapter 2).
Another reviewer's comment concerned the formula (67) and conclusions about the nonisotropic behavior of the material. It should be noted that the Termar-ADF composite friction material refers to the class of orthotropic materials with a more pronounced difference between thermal conductivity in the axial and radial directions. It is related to the structure of material. There are bundles arranged on the planes parallel to the heated surface of the disc and containing fibers embedded in the matrix. To perform calculations based on the linear model, we needed two constants, namely the thermal conductivity coefficients in the indicated directions. These are the effective constants, which takes into account the appropriate values for fiber and matrix and their geometric parameters. The methodology for determining effective thermal conductivity coefficients for such a composite is proposed in the monograph [38]. This is an approximate method, involving the linearization of heat flux lines passing through the unit cell of composite. Linearization is done by dividing the cell with a system of adiabatic or isothermal surfaces (or their combination), appropriately oriented relative to the general direction of the heat flux. In the article [29] such a division was made only by adiabatic planes, which, as indicated in monograph [38], is insufficient in determining the effective thermal conductivity coefficients. In our study, divisions were made by means of both adiabatic and isothermal surfaces, and the effective coefficients were determined as the arithmetic means of each of these division methods. The effective thermal conductivity coefficients determined using this methodology were adapted to the exact solutions obtained, and then the temperature evolution on the surface of the disc found on their basis was compared with the relevant experimental data [28] (Fig. 9).
In the proposed model, the effective thermal conductivity coefficient in radial direction is included in the parameter h* (8). From the limit Kr --> 0, we obtain h* --> 0 . This means that we have a model with one constant Kz (isotropic), but for adiabatic side surfaces of the disc. However, the relevant experimental data available from the article presented in Fig. 9 were made with forced cooling of the system. Unfortunately, we cannot make a Reviewer's recommendation based on this model.
For our justification it should be noted that Termar-ADF material is not publicly available material. For this reason, we did not have the opportunity to perform our own experimental test, but instead we used only data published in the literature.